# Orthogonal Random Features: Explicit Forms and Sharp Inequalities

**Nizar Demni**                                                  *nizar.demni@univ-amu.fr*
*Department of Mathematics*
*Aix-Marseille University, CNRS, LIS*
*Marseille, France*

**Hachem Kadri**                                                 *hachem.kadri@univ-amu.fr*
*Department of Computer Science*
*Aix-Marseille University, CNRS, LIS*
*Marseille, France*

**Reviewed on OpenReview:** *https://openreview.net/forum?id=FMtRZ4xzSi*

## Abstract

Random features have been introduced to scale up kernel methods via randomization techniques. In particular, random Fourier features and orthogonal random features were used to approximate the popular Gaussian kernel. Random Fourier features are built in this case using a random Gaussian matrix. In this work, we analyze the bias and the variance of the kernel approximation based on orthogonal random features which makes use of Haar orthogonal matrices. We provide explicit expressions for these quantities using normalized Bessel functions, showing that orthogonal random features does not approximate the Gaussian kernel but a Bessel kernel. We also derive sharp exponential bounds supporting the view that orthogonal random features are less dispersed than random Fourier features.

## 1 Introduction

Since their introduction over fifteen years ago in the seminal paper by Rahimi & Recht (2007), random features have become an important subject of research in the field of machine learning (see the review article by Liu et al., 2021). The primary motivation behind introducing them is to reduce the computation and storage requirements of kernel methods—one of the most popular machine learning approaches (Schölkopf & Smola, 2002; Shawe-Taylor & Cristianini, 2004; Yang et al., 2012; Le et al., 2013; Pennington et al., 2015; Chamakh et al., 2020; Han et al., 2022; Likhosherstov et al., 2022). They have also been used in over-parameterized settings and as a tool for generating and testing hypotheses on the generalization of deep learning (Jacot et al., 2018; Belkin et al., 2019; Yehudai & Shamir, 2019; Jacot et al., 2020; Liu et al., 2022; Mei & Montanari, 2022). Another recent research focus has involved the study of the Gaussian equivalence phenomenon in the context of random feature models in order to characterize the generalization error in the asymptotic regime. (Gerace et al., 2020; Goldt et al., 2022; Hu & Lu, 2022; Montanari & Saeed, 2022; Schröder et al., 2023; Dandi et al., 2024).

Random Fourier features (RFF) are undoubtedly the most common and widely used random feature method for kernel approximation (Rahimi & Recht, 2007). This approach applies to radial basis function kernels—a large class of kernel functions. It is based on Bochner's theorem (Bochner, 1932; Rudin, 1962), which establishes a one-to-one correspondence between continuous positive-definite functions and the Fourier transform of probability measures. In particular, if $\varphi$ is a real positive definite function on $\mathbb{R}$, i.e. the inequality $\sum_{i,j=1}^{n} \alpha_i \alpha_j \varphi(x_i - x_j) \geq 0$ holds for any $n \geq 1$ and $x_1, \ldots, x_n, \alpha_1, \ldots, \alpha_n \in \mathbb{R}$, and if $k(x,y) := \varphi(\|x - y\|)$

is a translation-invariant and radial kernel on $\mathbb{R}^d \times \mathbb{R}^d$, then there exists a non-negative measure $\mu$ such that

$$k(x, y) := \varphi(\|x - y\|) = \int_{\mathbb{R}^d} \exp(i w^\top (x - y)) d\mu(w), \tag{1}$$

where $\top$ stands for the transpose operation, and $\mu$ is invariant under orthogonal transformations. RFF consist in approximating the kernel $k$ by the following one defined by:

$$\tilde{k}(x, y) := \tilde{\phi}(x)^\top \tilde{\phi}(y), \quad \forall x, y, \in \mathbb{R}^d, \tag{2}$$

where

$$\tilde{\phi}(x) := \frac{1}{\sqrt{p}} (\sin(w_1^\top x), \ldots, \sin(w_p^\top x), \cos(w_1^\top x), \ldots, \cos(w_p^\top x))^\top, \tag{3}$$

and $w_1, \ldots, w_p \in \mathbb{R}^d$ are sampled from the distribution $\mu$. Indeed, one has

$$\tilde{\phi}(x)^\top \tilde{\phi}(y) = \frac{1}{p} \sum_{j=1}^p \cos(w_j^\top (x - y)),$$

so that the expected value of $\tilde{k}(x, y)$ fits exactly the integral displayed in the RHS of equation 1. The imaginary part of the integral in equation 1 vanishes because the kernel function is radial. In particular, if $w_1, \ldots, w_p$ are centered Gaussian vectors $\mathcal{N}(0, \mathrm{Id})$, then RFF approximate the well-known Gaussian kernel:

$$\mathbb{E}_{w \sim \mathcal{N}(0, \mathrm{Id})}[\tilde{k}(x, y)] = e^{-\|x - y\|^2 / 2}. \tag{4}$$

Orthogonal random features (ORF)[1] is a variant of RFF that uses a random orthogonal matrix $O$ instead of the Gaussian matrix $W$. RFF are built out of i.i.d. random Gaussian matrices. Assuming i.i.d. feature transformation samples may be too restrictive and may neglect structural information in the data. ORF might be a more effective model since it takes into accounts the correlations between samples. The feature transformation samples in ORF are no longer independent as they are drawn from the Haar measure on the orthogonal group. ORF was first proposed by Yu et al. (2016), with the goal of approximating the Gaussian kernel. They showed that imposing orthogonality on the randomly generated transformation matrix can reduce the kernel approximation error of RFF when a Gaussian kernel is used. This has led to a number of studies investigating the effectiveness of random orthogonal embeddings and showing empirically and theoretically that ORF estimators achieve more accurate kernel approximation and better prediction accuracy than standard mechanisms based on i.i.d sampling (Choromanski et al., 2017; 2018; 2019). The superiority of orthogonal against random Gaussian projections in learning with random features has also been observed in Gerace et al. (2020).

In this paper, we build on this line of research and provide an analytic characterization of the bias and of the variance of ORF using normalized Bessel functions of the first kind. These special functions appear naturally in harmonic analysis on Euclidean spheres since they are Fourier transforms of uniform measures on these spaces (Watson, 1995). Specifically, we make the following contributions:

- We give explicit forms of the bias and of the variance of ORF in the case where the random orthogonal matrix $O$ is drawn from the Haar measure. In particular, we show that the bias of ORF is given by a Bessel kernel instead of a Gaussian one.

- We derive sharp exponential bounds for these two quantities that are much tighter than the already known ones.

- We prove that the variance of ORF is less than the one of RFF in an interval whose length grows linearly with the data dimension.

---

[1]The model we consider here is the one denoted ORF' in Yu et al. (2016), Section 4, and is an instance of Definition 2.2. in Choromanski et al. (2018) corresponding to one block of features.

- We corroborate our theoretical findings with numerical experiments, supporting previous works showing the beneficial effect of orthogonality on random features.

The rest of the paper is organized as follows. Section 2 lays out notations needed for the statement of our main results, the latter being subsequently presented in Section 3. Numerical validation of them is provided in Section 4, while Section 5 concludes the paper. Proofs of all results are deferred to appendices.

## 2  Notation and preliminaries

Let $p, d$, be positive integers such that $2 \leq p \leq d$ and take a Haar $p \times p$ orthogonal matrix $O$. For the reader's convenience, recall that the Haar measure on the orthogonal group is the unique left and right invariant measure and that a Haar orthogonal matrix may be obtained using the Gram-Schmidt procedure applied to a Gaussian matrix $G$. In particular, the columns (and rows) of $O$ are uniformly distributed on the sphere (for further details see Meckes 2019, Chapter 1). It is worth noting that the value $p = 1$ is excluded since trivial. If $p > d$, then the ORF feature map, as introduced in Choromanski et al. (2018), is built out of independent blocks of vectors sampled from independent Haar orthogonal matrices. By linearity and statistical independence, the computations of the bias and of the variance of the ORF estimator follows in this case from summing those we will compute below.

We denote by $\tilde{\phi}_{ORF}$ the random features of ORF computed using equation 3 with $w_1, \ldots, w_p$ being columns of $O$. We also use the notation $\tilde{\phi}_{RFF}$ for the random features of RFF when $w_1, \ldots, w_p$ are columns of a Gaussian matrix $G$ (i.e., a random matrix whose entries are independent and centered Gaussian random variables). The approximate kernels obtained using ORF and RFF will be denoted by $\tilde{k}_{ORF}(x, y)$ and $\tilde{k}_{RFF}(x, y)$, respectively (i.e., $\tilde{k}_{ORF}(x, y) := \tilde{\phi}_{ORF}(x)^\top \tilde{\phi}_{ORF}(y)$ and $\tilde{k}_{RFF}(x, y) := \tilde{\phi}_{RFF}(x)^\top \tilde{\phi}_{RFF}(y)$). In order to simplify the exposition and without loss of generality, we will assume throughout this paper that the bandwidth of the Gaussian kernel $\sigma$ is equal to 1. In this respect and for sake of completeness, let us recall the expressions of the bias and the variance of RFF.

**Theorem 1** (Bias and variance of $\tilde{\mathbf{k}}_{\mathbf{RFF}}(\mathbf{x}, \mathbf{y})$). *Let $\tilde{k}_{RFF}(x, y)$ be the RFF-based approximate kernel computed with $p$ random vectors in $\mathbb{R}^d$. Then its expectation and its variance are given by*

$$\mathbb{E}[\tilde{k}_{RFF}(x, y)] = \exp\left(-\frac{\|x - y\|^2}{2}\right), \tag{5}$$

*and*

$$V[\tilde{k}_{RFF}(x, y)] = \frac{1}{2p}\left(1 - \exp\left(-\frac{\|x - y\|^2}{2}\right)\right)^2, \tag{6}$$

*respectively.*

*Proof.* See Yu et al. (2016, Lemma 1). □

It is worth noting that the equality equation 5 remains valid if one replaces the Gaussian matrix $G$ by the product $SO$, where $O$ is a Haar orthogonal matrix and $S$ is a diagonal matrix whose entries are independent and $\chi_2$-distributed random variables with $d$ degrees of freedom (Yu et al., 2016, Theorem 1). However, it fails when $w_1, \ldots, w_p$ are columns of $O$. We will see in the next section how the bias and the variance change and behave in this case.

## 3  Main results

In this section, we state the main results of this paper and comment on our findings. We start with an explicit expression of the bias of ORF-based kernel approximation.

**Theorem 2** (Bias of $\tilde{\mathbf{k}}_{\mathbf{ORF}}(\mathbf{x}, \mathbf{y})$). *Let $\tilde{k}_{ORF}(x, y)$ be the ORF-based approximate kernel computed with $p$ random vectors in $\mathbb{R}^d$. Then its expectation reads:*

$$\mathbb{E}[\tilde{k}_{ORF}(x, y)] = j_{d/2-1}(z), \quad z := \|x - y\|, \tag{7}$$

where $j_{d/2-1}(\cdot)$ is the normalized Bessel function of the first kind defined by:

$$j_{d/2-1}(z) = \sum_{n \geq 0} \frac{(-1)^n \Gamma(d/2)}{n! \Gamma(n+(d/2))} \left(\frac{z}{2}\right)^{2n}, \tag{8}$$

with $\Gamma(\cdot)$ being the Gamma function.

*Sktech of proof.* The proof of Theorem 2 is a routine calculation in Euclidean harmonic analysis. Loosely speaking, it relies on the invariance under rotations of the uniform (Haar) measure on the unit sphere $S^{d-1}$ which allows to reduce the expectation $\mathbb{E}[\tilde{k}_{ORF}(x,y)]$ to the one-dimensional Fourier transform of the symmetric Beta distribution whose density is proportional to

$$(1-u^2)^{(d-3)/2}, \quad u \in [-1,1].$$

The detailed proof is available in Appendix A. □

Theorem 2 shows that ORF approximate the kernel defined by the normalized Bessel function of the first kind (Watson, 1995; Shishkina & Sitnik, 2020, Chapter 1). This function is oscillating in contrast to the so-called Matern kernel given by the modified Bessel function of the second kind (see Equation 12 in Genton 2001). Moreover, the absolute value of the former admits a polynomial decay to zero as its argument becomes large while the latter decays exponentially.

To address the question of how the bias of ORF behaves compared to that of RFF, we use Theorem 2 to prove the following result.

**Proposition 3.** *For all $x, y \in \mathbb{R}^d$, let $z := \|x - y\|$ and define*

$$b_d := 2^{1/4} d^{3/4} \sqrt{1 - \frac{4}{2\sqrt{2}d^{3/2} - d}}, \quad d \geq 2,$$

$$c_d := \left(\frac{d^2}{4} - 1\right)^{1/2} \sqrt{1 - \frac{8}{d^2 - 2d - 4}}, \quad d \geq 5.$$

*Then for all $z \in [0, \max(b_d, c_d)]$, we have*

$$e^{-z^2/2} \leq \mathbb{E}[\tilde{k}_{ORF}(x,y)] \leq e^{-z^2/(2d)}, \tag{9}$$

*where we convent that $\max(b_d, c_d) = b_d$ when $2 \leq d \leq 4$. The upper bound is valid up to the second zero of $j_{d/2-1}$.*

*Sketch of proof.* The series expansion equation 8 is clearly sign alternating and does not help in proving inequalities for Bessel functions of the first kind by killing oscillations. For that reason, we appeal to the Weierstrass infinite product of $j_{(d-2)/2}$ and make use of estimates of its smallest positive zero. Though the Bessel function $j_{(d-2)/2}$ admits infinitely many simple zeros, the estimates on the smallest one are sufficient enough for our purposes. The detailed proof is provided in Appendix B. □

For fixed $d \geq 5$, the constants $b_d$ and $c_d$ are the values of the increasing function

$$f_d : u \mapsto u\sqrt{1 - \frac{4}{2u^2 - d}}, \quad u \geq \frac{d+4}{2},$$

at two lower bounds of the first zero of $j_{(d/2)-1}$; see eqs. equation 18 and equation 19 below. Moreover, $c_d > b_d$ for sufficiently large $d$ ($d \gtrsim 35$) and offers therefore a linear growth of the interval $[0, c_d]$ compared to $d^{3/4}$ for $[0, b_d]$. As a matter of fact, the inequalities displayed in equation 9 hold true for a large range of $z$ provided that $d$ is large as well. As we shall see later from the proof of proposition equation 3, the lower bound even holds true in a larger interval which almost reaches the first zero of $j_{(d/2)-1}$ as $d$ becomes large.

On the other hand, the Bessel function $j_{(d/2)-1}$ takes negative values after vanishing at its first zero so that our lower bound is quite sharp.

As to the upper bound, it clearly remains valid up to the second zero of $j_{(d/2)-1}$ since the latter is non positive between its first and its second zeroes. Note also that the lower and the upper bounds correspond to Gaussian kernels with standard deviations equal to one and to $\sqrt{d}$ respectively.

On the other hand, we may equivalently write

$$0 \leq \mathbb{E}[\tilde{k}_{ORF}(x,y)] - \mathbb{E}[\tilde{k}_{RFF}(x,y)] \leq e^{-z^2/(2d)} - e^{-z^2/2}, \quad \forall z \in [0, \max(b_d, c_d)], \tag{10}$$

which considerably improves Theorem 2 in Yu et al. (2016). Indeed, our upper bound decays exponentially fast while the one given in Yu et al. (2016) admits an exponential growth. This growth is due to the fact that the triangular inequality used in the proof there kills the oscillations of the normalized Bessel function $j_{d/2-1}$ and leads to the normalized modified Bessel function of the first kind which is known to grow exponentially (Watson, 1995).

It is also worth mentioning that for large enough $d$, the differences between $\mathbb{E}[\tilde{k}_{ORF}(x,y)]$ and the exponential bounds displayed in equation 9 become very small for large $z$ ($z \gg d$) since $|j_{(d/2)-1}|$ has a polynomial decay to zero.

We now state our second main result providing an explicit closed expression of the variance of ORF by means of normalized Bessel functions of the first kind.

**Theorem 4** (Variance of $\tilde{k}_{ORF}(\mathbf{x}, \mathbf{y})$). *Let $\tilde{k}_{ORF}(x,y)$ be the ORF-based approximate kernel built out of $p$ random vectors in $\mathbb{R}^d$. Then its variance is given by:*

$$V[\tilde{k}_{ORF}(x,y)] = \frac{1}{p} \left\{ \frac{1 + j_{(d/2)-1}(2z)}{2} + (p-1)j_{(d/2)-1}(\sqrt{2}z) - p \left( j_{(d/2)-1}(z) \right)^2 \right\}, \tag{11}$$

*where we recall the notation $z = \|x - y\|$.*

*Sketch of proof.* The derivation of $V[\tilde{k}_{ORF}(x,y)]$ is to the best of our knowledge new. Since $\tilde{k}_{ORF}(x,y)$ is the sum of correlated random variables in opposite to the RFF kernel, one has to compute explicitly the covariance terms which we perform by exploiting once more the invariance of the uniform measure on $S^{d-1}$ which leads to a double integral that we evaluate using polar coordinates. The full details of the proof can be found in Appendix C. □

**Remark 5.** *When $p = 1$, the variance reduces to*

$$V\left(K(x,y)\right) = \frac{1 + j_{(d-2)/2}(2z)}{2} - \left( j_{(d-2)/2}(z) \right)^2.$$

*The non negativity of the RHS follows also from the inequality:*

$$[j_\nu(x) + j_\nu(y)]^2 \leq [1 + j_\nu(x+y)][1 + j_\nu(x-y)]$$

*valid for any $\nu > -1/2$ and any $x, y \in \mathbb{R}$ (Neuman, 2004).*

Using equation 9, this variance can be bounded as follows for all $z \in [0, \max(b_d, c_d)]$:

$$\frac{1 + e^{-2z^2}}{2p} + \frac{p-1}{p}e^{-z^2} - e^{-z^2/d} \leq V[\tilde{k}_{ORF}(x,y)] \leq \frac{1 + e^{-2z^2/d}}{2p} + \frac{p-1}{p}e^{-z^2/d} - e^{-z^2}. \tag{12}$$

Similarly to equation 10, the upper bound in equation 12 is much sharper than the one in Yu et al. (2016, Theorem 2). However, it does not give information about how the variance of ORF compares to that of RFF. The following proposition provides an answer to this question.

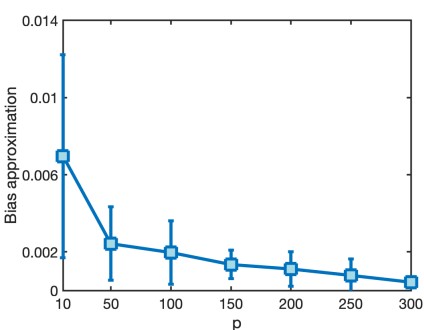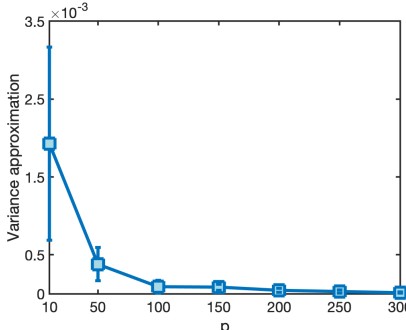

Figure 1: The absolute difference between theoretical and empirical values of the bias and the variance of ORF for different values of the number of random features $p$. **Left:** $\left|M_{emp} - \mathbb{E}[\tilde{k}_{ORF}(x,y)]\right|$. **Right:** $\left|V_{emp} - V[\tilde{k}_{ORF}(x,y)]\right|$. The bias and variance of $\tilde{k}_{ORF}(x,y)$, $\mathbb{E}[\tilde{k}_{ORF}(x,y)]$ and $V[\tilde{k}_{ORF}(x,y)]$, are computed using the explicit closed expressions provided in Theorems 2 and 4. $M_{emp}$ and $V_{emp}$ are the empirical bias and variance, respectively. Data points $x$ and $y$ are randomly generated from a normal distribution with zero mean and unit variance. We consider here the case where the value of $z := \|x - y\|$ is not small ($z$ is equal to 24 in this simulation).

**Proposition 6.** *For $d \geq 2$, denote*

$$\alpha_d \;\; := \;\; \left(\frac{d}{2}\right)^{3/4},$$

$$\beta_d \;\; := \;\; \frac{1}{2}\sqrt{\frac{d^2}{4} - 1}.$$

*Then for any $x, y \in \mathbb{R}^d$ and any $z = \|x - y\| \in [0, \max(\alpha_d, \beta_d)]$, we have*

$$V[\tilde{k}_{ORF}(x,y)] \leq V[\tilde{k}_{RFF}(x,y)]. \tag{13}$$

*Sketch of proof.* Since our random features are not independent, one has to focus on the covariance terms. Appealing to the infinite product representation of the spherical Bessel function $j_{(d/2)-1}$, it turns out that these terms are negative. As such, we are led to analyse the variance of a single random feature which is proved to be less than the variance of RFF in the indicated interval. The complete proof is given in Appendix D. □

Proposition 6 shows that $\tilde{k}_{ORF}$ is less dispersed than $\tilde{k}_{RFF}$ when the norm difference between data points $z$ lies within an interval whose length is linear in the data dimension $d$ when the latter is sufficiently large. This is in agreement with previous results (Choromanski et al., 2017; 2018; 2019) though holding in a small $z$-neighborhood of zero.

Another striking feature of this proposition is its independence of the number of random features $p$. Actually, its proof relies again on the Weierstrass infinite product of $j_{(d/2)-1}$ and shows in particular that the covariance of $(\cos(w_i^T(x-y)), \cos(w_j^T(x-y)), i \neq j$, is negative in the interval $[0, \sqrt{2}\max(\alpha_d, \beta_d)]$. As a matter of fact, one is left with bounding from above the variance of a single mode $\cos(w_1^T(x-y))$ by that of the RFF kernel. Even more, our proof shows that the inequality equation 13 remains valid only in a slightly larger interval where $j_{(d/2)-1}(\sqrt{2}z)$ is negative.

## 4 Numerical illustrations

In this section we provide experimental results on synthetic and real data that corroborate our theoretical findings.

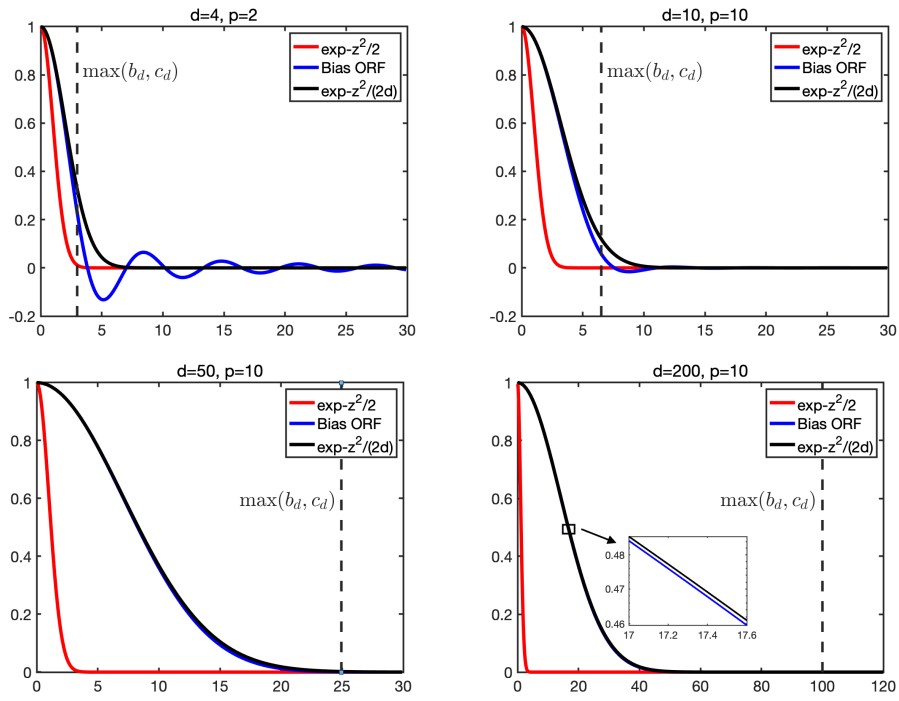

Figure 2: The bias of $\tilde{k}_{ORF}(x, y)$ and bounds of Proposition 3 as a function of $z := \|x - y\|$.

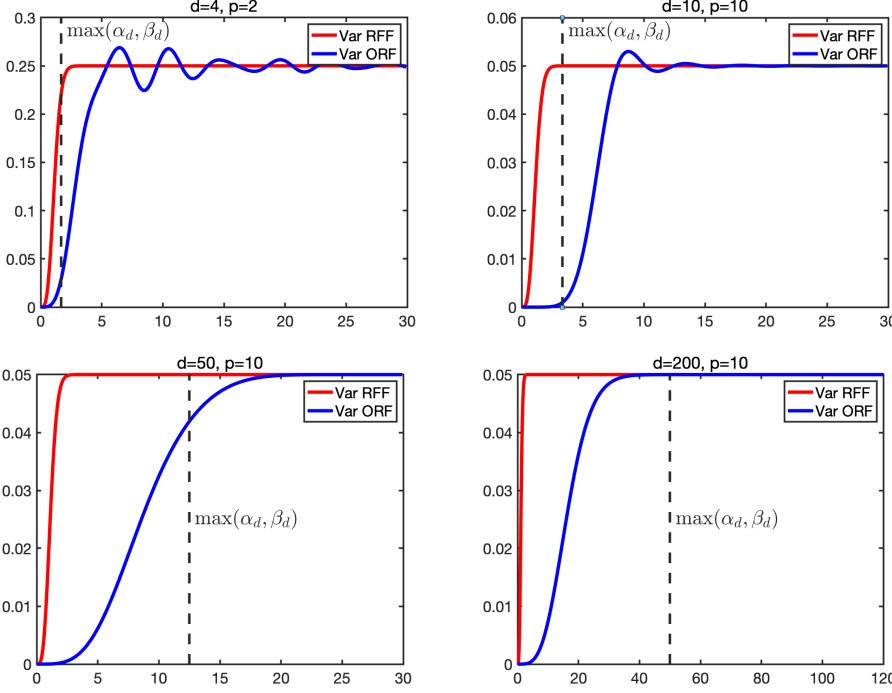

Figure 3: The variance of $\tilde{k}_{ORF}(x, y)$ and $\tilde{k}_{RFF}(x, y)$ as a function of $z := \|x - y\|$.

Table 1: Dataset statistics. $d$ is the dimension of the features.

| Datasets | $d$ | # data points |
|---|---|---|
| Ionosphere[2] | 34 | 351 |
| Ovariancancer[3] | 100 | 216 |
| Campaign[4] | 62 | 41,188 |
| Backdoor[5] | 196 | 95,329 |

## 4.1 Synthetic data results

We generate synthetic data with dimension $d = 300$ and varying values of the random features $p = \{10, 50, 100, 150, 200, 250, 300\}$. The data are randomly generated from a normal distribution with zero mean and unit variance. We compute $M_{emp} := \frac{1}{s}\sum_{l=1}^{s}\tilde{k}_l(x,y)$ and $V_{emp} := \frac{1}{s}\sum_{l=1}^{s}(\tilde{k}_l(x,y) - M_{emp})^2$, the empirical bias and variance of $\tilde{k}_{ORF}$ respectively. Each kernel $\tilde{k}_l$ is computed using a random Haar orthogonal matrix $O^l$, i.e., $\tilde{k}_l(x,y) = \tilde{\phi}_l(x)^\top\tilde{\phi}_l(y)$ where $\tilde{\phi}_l(x) = \frac{1}{\sqrt{p}}\big(\sin(w_1^{l\top}x),\ldots,\sin(w_p^{l\top}x),\cos(w_1^{l\top}x),\ldots,\cos(w_p^{l\top}x)\big)^\top$ and $w_1^l,\ldots,w_p^l$ are the columns of $O^l$. The experiment is repeated 10 times with different random seeds. Figure 1 shows the approximation errors $\big|M_{emp} - \mathbb{E}[\tilde{k}_{ORF}(x,y)]\big|$ and $\big|V_{emp} - V[\tilde{k}_{ORF}(x,y)]\big|$ for $s = 50$ and for different values of $p$. The mean and variance of $\tilde{k}_{ORF}(x,y)$, $\mathbb{E}[\tilde{k}_{ORF}(x,y)]$ and $V[\tilde{k}_{ORF}(x,y)]$, are computed using the explicit closed expressions provided in Theorems 2 and 4. As can be seen, the mean and variance approximation errors are very small, which are in agreement with our results.

Figure 2 shows the bias of $\tilde{k}_{ORF}(x,y)$ and the bounds of Proposition 3 as a function of $z = \|x - y\|$. It illustrates that inequalities in equation 9 hold for any $z \in [0, \max(b_d, c_d)]$. Figure 3 depicts the variance of $\tilde{k}_{ORF}$ and $\tilde{k}_{RFF}$. It confirms that ORF has smaller variance compared to the standard RFF, as claimed in Proposition 6.

## 4.2 Real data results

We also conduct experiments on real-world datasets to confirm our theoretical findings. The number of feature dimension and data samples for each dataset are provided in Table 1. The accuracy of the kernel estimation is calculated by measuring the mean squared error (MSE) between the true kernel matrix and the approximated one. Figure 4 compares the MSE of ORF and RFF, i.e., $\|K - \tilde{K}\|_F^2/n^2$ where $K := [k(x_i, x_j)]_{i,j=1}^n$ is the Bessel or Gaussian kernel matrix and $\tilde{K}$ is its approximation via ORF or RFF, respectively. The Gaussian kernel bandwidth $\sigma$ is set as the average distance between all pairs of data points, i.e., $\sigma = \sqrt{1/n^2\sum_{i,j=1}^n\|x_i - x_j\|^2}$. For the ORF estimator, introducing such a bandwidth is somehow artificial since the uniform measure on the sphere may be realized as a normalized Gaussian vector (so even if we start with a Gaussian vector whose standard deviation is $\sigma$, this parameter disappears after normalization). The experiment is repeated five times with different random seeds. ORF often achieves lower MSE than RFF. Note that the MSE measures the quadratic variability with respect to the empirical data between the estimator and its theoretical mean (the latter is taken with respect to the random features). In other words, the MSE corresponds to an empirical approximation of the variance. Figure 4 supports Proposition 6 since it shows that for the same number of feature $p$, the approximation error of the kernel function (i.e., empirical variance) in the ORF setting is smaller than the one in the RFF setting.

---

[3]Ionosphere data from the UCI machine learning repository: `https://archive.ics.uci.edu/dataset/52/ionosphere`.

[4]Ovarian cancer data (Conrads et al., 2004): `https://fr.mathworks.com/help/stats/sample-data-sets.html`.

[5]Campaign data is a data set of direct bank marketing campaigns via phone calls (Pang et al., 2019): `https://github.com/GuansongPang/ADRepository-Anomaly-detection-datasets#numerical-datasets`.

[6]Backdoor attack detection data extracted from the UNSW-NB 15 dataset (Moustafa & Slay, 2015): `https://github.com/GuansongPang/ADRepository-Anomaly-detection-datasets#numerical-datasets`.

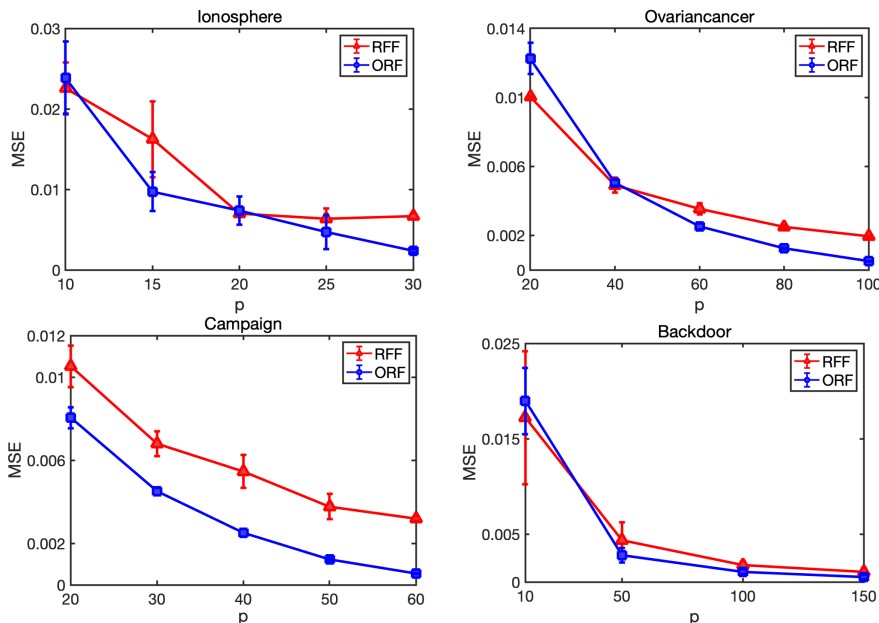

Figure 4: Mean squared error (MSE) between the kernel matrix approximated by ORF or RFF and the full kernel matrix computed by the Bessel or the Gaussian kernel, for different values of the number of random features $p$.

## 5 Conclusion

In this paper, we provided explicit closed expressions of the bias and of the variance of ORF by means of normalized Bessel functions of the first kind. We also derived exponential bounds that improve previously known ones. In particular, we proved that the variance of ORF is less than the one of RFF when the norm difference between data points lies in an interval of length $O(d)$, $d$ being the data dimension.

### Acknowledgments

We thank the reviewers for their helpful comments and suggestions. This work has been funded by the French National Research Agency (ANR) under Grant No. ANR-19-CE23-0011 (project QuantML).

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

## A Proof of Theorem 2

**Proof**. By linearity of the expectation and since all the vectors $w_j, 1 \leq j \leq p$, are uniformly distributed on the sphere, we obviously have:

$$\mathbb{E}[\tilde{k}_{ORF}(x,y)] = \mathbb{E}[\cos(w_1^T(x-y)].$$

Moreover, we can find an orthogonal matrix $O_{x,y}$ such that

$$O_{x,y}(x-y) = ||x-y||e_1,$$

where $e_1$ is the first vector of the canonical basis of $\mathbb{R}^d$. Indeed, the columns of the transpose matrix $O_{x,y}^T$ consists of the normalized vector $(x-y)/||x-y||$ together with any set of vectors forming an orthonormal basis of $\mathbb{R}^d$.

Now, since the uniform measure on the sphere is invariant by orthogonal transformation, we further get:

$$\mathbb{E}[\tilde{k}_{ORF}(x,y)] = \mathbb{E}[\cos(w_1^T O_{x,y}(x-y)]$$
$$= \mathbb{E}[\cos(||x-y||w_{11})],$$

where $w_{11}$ is the first coordinate of $w_1$. This real random variable follows the beta distribution whose density is given by:

$$\frac{\Gamma(d/2)}{\sqrt{\pi}\Gamma((d-1)/2)}(1-u^2)^{(d-3)/2}, \quad -1 < u < 1.$$

Theorem 2 follows from the Poisson integral representation of the normalized Bessel function of the first kind (Watson (1995), Ch. II):

$$j_\nu(z) = \frac{\Gamma(\nu+1)}{\sqrt{\pi}\Gamma(\nu+(1/2))}\int_{-1}^1 \cos(zu)(1-u^2)^{\nu-1/2}, \quad \nu > -1/2,$$

which may be derived by expanding the cosine function in the right-hand side into power series and integrating termwise.

## B Proof of Proposition 3

**Proof** As briefly sketched right after the statement of Proposition 3, we shall make use of the infinite product representation of The normalized Bessel function $j_{d/2-1}$ recalled below. To this end, recall from Watson (1995) that this function admits an infinite number of positive simple zeros increasing to infinity:

$$0 < a_{d,1} < a_{d,2} < \ldots$$

As a matter of fact, one has (Watson 1995, p.498):

$$j_{(d/2)-1}(z) = \prod_{j=1}^\infty \left(1 - \frac{z^2}{(a_{d,j})^2}\right). \tag{14}$$

Now, in order to prove the upper bound, we further use the so-called first Rayleigh sum (Watson, 1995, p. 502):

$$\sum_{j\geq 1}\frac{1}{(a_{d,j})^2} = \frac{1}{2d}. \tag{15}$$

Indeed, the inequality $1 - u \leq e^{-u}, u \in [0,1]$ holds so that if $z \leq a_{d,1}$ then

$$j_{(d/2)-1}(z) \leq e^{-\sum_{j\geq 1} z^2/(a_{d,j})^2} = e^{-z^2/(2d)} \tag{16}$$

yielding the upper bound. Note that the latter remains true in the interval $[a_{d,1}, a_{d,2}]$ since the Bessel function is non positive there.

As to the derivation of the lower bound, it is more technical and relies on fine properties of Bessel functions. Firstly, We differentiate the function

$$h_d : z \mapsto e^{z^2/2} j_{(d/2)-1}(z), \quad 0 \leq z \leq a_{d,1},$$

and note using straightforward computations that

$$(j_{(d/2)-1})'(z) = -\frac{z}{d} j_{(d/2)}(z).$$

It follows that:

$$h_d'(z) = \frac{e^{z^2/2} j_{(d/2)-1}(z)}{d} \left( d - \frac{j_{(d/2)}(z)}{j_{(d/2)-1}(z)} \right).$$

Secondly, we appeal to the Mittag-Leffler expansion (Ifantis & Siafarikas, 1990, eq. 2.9):

$$\frac{j_{(d/2)}(z)}{j_{(d/2)-1}(z)} = 2d \sum_{m=1}^{\infty} \frac{1}{a_{d,m}^2 - z^2}$$

to infer that the equation $h_d'(z) = 0, 0 < z < a_{d,1}$, is equivalent to

$$\sum_{m=1}^{\infty} \frac{1}{a_{d,m}^2 - z^2} = 1/2.$$

But the LHS of the last equality is obviously increasing in the $z$-variable and tends to $+\infty$ as $z \to a_{d,1}$. As a result, the first Rayleigh sum equation 15 (giving the value of the above series at $z = 0$) implies the existence of one and only one solution $z_0(d)$ to the equation $h_d'(z) = 0$ in $(0, a_{d,1})$. Therefore,

$$h_d(z) \geq 1 \quad \Leftrightarrow \quad e^{-z^2/2} \leq j_{(d/2)-1}(z),$$

for any $z \in [0, z_0(d)]$.

It then remains to seek a more precise estimate of the real number $z_0(d)$. To this end, we appeal to the following inequality which readily follows from Theorem 2.1 in Freitas (2021):

$$\sum_{m=1}^{\infty} \frac{1}{a_{d,m}^2 - z^2} \leq \frac{1}{a_{d,1}^2 - z^2} + \frac{d}{4a_{d,1}^2},$$

to see that

$$\frac{1}{2} \leq \frac{1}{a_{d,1}^2 - [z_0(d)]^2} + \frac{d}{4a_{d,1}^2}.$$

Equivalently,

$$z_0(d) \geq a_{d,1} \sqrt{1 - \frac{4}{2a_{d,1}^2 - d}} = f_d(a_{d,1}). \tag{17}$$

The sought estimate follows then from the lower bounds below: (Ismail & Muldoon, 1988, eq. 5.4):

$$a_{d,1} > \sqrt{2d} \left( \frac{d}{2} \right)^{1/4} = 2^{1/4} d^{3/4}, \quad d \geq 2, \tag{18}$$

and (Watson, 1995, eq. 5, p. 486)

$$a_{d,1} > \sqrt{\frac{d^2}{4} - 1}, \quad d \geq 2. \tag{19}$$

Actually, $f_d$ is increasing for fixed $d$ so that

$$z_0(d) > \max \left( f(2^{1/4} d^{3/4}) = b_d, f\left( \frac{d^2}{4} - 1 \right) = c_d \right)$$

which completes the proof.

**Remark 7.** *If we use the following inequality (Joshi & Bissu, 1996, eq. 2.6):*

$$1 - \frac{z^2}{2d} \leq j_{(d/2)-1}(z),$$

*we can prove that the lower bound holds in the interval $[0, \sqrt{d}]$.*

## C  Proof of Theorem 4

**Proof** Let us consider the variance of $K(x, y)$:

$$
\begin{aligned}
V\left(\tilde{k}_{ORF}(x, y)\right) &= \frac{1}{p^2} V\left[\sum_{j=1}^{p} \cos(w_j^T(x - y))\right] \\
&= \frac{1}{p} V\left[\cos(w_1^T(x - y))\right] + \frac{p-1}{p} \operatorname{cov}\left[\cos(w_1^T(x - y)), \cos(w_2^T(x - y))\right].
\end{aligned}
$$

Now, the linearization formula for the cosine function

$$\cos^2(\theta) = \frac{1 + \cos(2\theta)}{2}$$

entails

$$
\begin{aligned}
V\left[\cos(w_1^T(x - y))\right] &= \mathbb{E}[\cos^2(w_1^T(x - y))] - \left[\mathbb{E}[\cos(w_1^T(x - y))]\right]^2 \\
&= \frac{1}{2} + \frac{j_{(d/2)-1}(2z)}{2} - \left(j_{(d/2)-1}(z)\right)^2.
\end{aligned}
$$

As to the covariance term, the invariance of the Haar distribution (keep in mind the orthogonal matrix $O_{x,y}$) together with the product formula

$$\cos(a)\cos(b) = \frac{1}{2}[\cos(a + b) + \cos(a - b)]$$

yield:

$$
\begin{aligned}
\operatorname{cov}\left[\cos(w_1^T(x - y)), \cos(w_2^T(x - y))\right] &= \mathbb{E}[\cos(w_1^T(x - y))\cos(w_2^T(x - y))] - \left(j_{(d-2)/2}(z)\right)^2 \\
&= \mathbb{E}[\cos(w_{11}z)\cos(w_{12}z)] - \left(j_{(d-2)/2}(z)\right)^2 \\
&= \frac{1}{2}\left\{\mathbb{E}[\cos((w_{11} + w_{12})z)] + \mathbb{E}[\cos((w_{11} - w_{12})z)]\right\} \\
&\quad - \left(j_{(d/2)-1}(z)\right)^2
\end{aligned}
$$

where $w_{11}$ and $w_{12}$ are the first coordinates of the column vectors $w_1$ and $w_2$. But $(w_{11}, w_{12})$ is the first row of the Haar orthogonal matrix $O$ which is uniformly distributed on $S^{d-1}$. Consequently, the joint distribution of $(w_{11}, w_{12})$ is given by the following probability density:

$$\frac{d-2}{2\pi}(1 - u^2 - v^2)^{(d/2)-2}\mathbf{1}_{\{u^2+v^2<1\}}$$

with respect to Lebesgue measure $du\, dv$, whence

$$\operatorname{cov}\left[\cos(w_1^T(x - y)), \cos(w_2^T(x - y))\right] = \mathbb{E}[\cos((w_{11} + w_{12})z)] - \left(j_{(d/2)-1}(z)\right)^2.$$

Moving to polar coordinates:

$$w_{11} = r\cos(\theta), \quad w_{12} = r\sin(\theta),$$

it follows that

$$\mathbb{E}[\cos((w_{11} + w_{12})z)] = \frac{d-2}{2\pi} \int_0^1 \int_0^{2\pi} [\cos(\cos\theta + \sin\theta)rz]r(1 - r^2)^{(d/2)-2}drd\theta. \tag{20}$$

Expanding further the cosine into power series, we are left with the following two integrals:

$$\int_0^1 r^{2j+1}(1 - r^2)^{(d/2)-2}dr = \frac{\Gamma(j+1)\Gamma((d/2)-1)}{2\Gamma(j+(d/2))}, \quad j \geq 0, \tag{21}$$

and (equation 3.66.1.2., p. 405 in Gradshteyn & Ryzhik, 2014):

$$\int_0^{2\pi} (\cos\theta + \sin\theta)^{2j}d\theta = 2\pi\frac{2^j(2j-1)!!}{(2j)!!}$$

where $(2j)!! = (2j)(2j-2)\cdots(2)$ is the double factorial and likewise $(2j-1)!! = (2j-1)(2j-3)\cdots(3)(1)$. Writing

$$(2j)!! = 2^j j!, \quad (2j-1)!! = \frac{(2j)!}{2^j j!},$$

we equivalently get:

$$\int_0^{2\pi} (\cos\theta + \sin\theta)^{2j}d\theta = 2\pi\frac{(2j)!}{2^j(j!)^2}. \tag{22}$$

Gathering equation 20, equation 21 and equation 22, we end up with the expression:

$$\mathbb{E}[\cos((w_{11} + w_{12})z)] = (d-2)\Gamma((d-2)/2)\sum_{j\geq 0}\frac{(-1)^j z^{2j}}{2^j j!\Gamma(j+(d/2))}$$

$$= j_{(d/2)-1}(\sqrt{2}z),$$

where we used the formula $(d-2)\Gamma((d-2)/2) = 2\Gamma(d/2)$. Finally

$$V\left(\tilde{k}_{ORF}(x,y)\right) = \frac{1}{p}\left\{\frac{1 + j_{(d/2)-1}(2z)}{2} - \left(j_{(d/2)-1}(z)\right)^2\right\} + \frac{p-1}{p}\left\{j_{(d/2)-1}(\sqrt{2}z) - \left(j_{(d/2)-1}(z)\right)^2\right\}$$

$$= \frac{1}{p}\left\{\frac{1 + j_{(d/2)-1}(2z)}{2} + (p-1)j_{(d/2)-1}(\sqrt{2}z) - p\left(j_{(d/2)-1}(z)\right)^2\right\},$$

as desired.

## D  Proof of Proposition 6

**Proof** The infinite product equation 14 and the inequality

$$1 - 2u \leq (1-u)^2, \quad u \geq 0,$$

implies that the covariance term computed above is non positive:

$$(p-1)\left[j_{(d/2)-1}(\sqrt{2}z) - \left(j_{(d/2)-1}(z)\right)^2\right] \leq 0$$

on the interval $[0, a_{d,1}/\sqrt{2}]$. Consequently,

$$V\left(K(x,y)\right) \leq \frac{1}{p}\left[\frac{1 + j_{(d/2)-1}(2z)}{2} - \left(j_{(d/2)-1}(z)\right)^2\right].$$

Similarly,

$$(1 - 4u) \leq (1-u)^4, \quad u \geq 0,$$

holds since the discriminant of
$$u^2 - 4u + 6, \quad u \geq 0,$$
is negative. Applying this inequality to each factor in the product equation 14 entails:
$$j_{(d/2)-1}(2z) \leq \left[j_{(d/2)-1}(z)\right]^4$$
on the interval $[0, a_{d,1}/2]$. Keeping in mind the lower bound derived in Proposition 3 which remains valid in $[0, z_0(d)]$ (see the proof), we get on the interval $[0, \inf(z_0(d), a_{d,1}/2)]$

$$
\begin{aligned}
pV\left(K(x,y)\right) &\leq \frac{1 + j_{(d/2)-1}(2z)}{2} - \left(j_{(d/2)-1}(z)\right)^2 \leq \frac{1 + \left[j_{(d/2)-1}(z)\right]^4 - 2\left(j_{(d/2)-1}(z)\right)^2}{2} \\
&= \frac{\left[1 - \left(j_{(d/2)-1}(z)\right)^2\right]^2}{2} \\
&\leq \frac{[1 - e^{-z^2}]^2}{2},
\end{aligned}
$$

giving the inequality stated in the proposition.

Finally, it remains to compare $z_0(d)$ and $a_{d,1}/2$. To proceed, note that by the virtue of equation 17, we are led to compare
$$\sqrt{1 - \frac{4}{2a_{d,1}^2 - d}}$$
and the value $1/2$. In this respect, the lower bound given in Ismail & Muldoon (1988), eq. (5.4), shows that

$$\sqrt{1 - \frac{4}{2a_{d,1}^2 - d}} > 1/2,$$

whence we infer that $z_0(d) > a_{d,1}/2$. Consequently, the inequality $V[\tilde{k}_{ORF}(x,y)] \leq V[\tilde{k}_{RFF}(x,y)]$ holds true for any $z = \|x - y\| \in [0, a_{d,1}/2]$. Recalling equation 18 and equation 19, we are done.

