# OpenReview forum: "Orthogonal Random Features: Explicit Forms and Sharp Inequalities"
_TMLR — Accepted by TMLR_

### Review · Reviewer_ZMow · 2024-06-05

**Summary Of Contributions:**

This manuscript characterizes the bias and variance associated with orthogonal random feature approximation of kernel matrices. The authors tackle the scenario in which the orthogonal matrices are drawn from the Haar measure. The main theoretical results consist of providing sharp bounds on bias and variance of this approximation scheme. The formal results are accompanied by numerical illustrations.

**Audience:**

Yes

**Broader Impact Concerns:**

I do not have any concerns about the ethical implications of the work.

**Claims And Evidence:**

Yes

**Requested Changes:**

My concerns regard mainly the readability of the main text, I list below different changes that I believe will enhance the presentation.


# Main concerns
- This manuscript nicely finds that, in contrast with the community wisdom, Orthogonal Random Feature approximates a different kernel w.r.t the Gaussian one. This is briefly mentioned in the abstract, but not emphasized enough in the introduction of the main results; more precisely, the authors should explain explicitly the common belief that ORF approximates the Gaussian kernel.
- Similarly to the above, the Orthogonal Random Feature approach is presented in five lines (Page 2) before listing the main theoretical contribution. The authors should justify in deeper detail (with proper references to related works) the important role that ORF plays. For example, “ORF are more accurate with respect to i.i.d sampling…” is an insufficient explanation; e.g., “more accurate” in what respect?
- It is difficult to navigate the writing of the main technical results. There are no proof sketches and they lack intuitive explanations that would help to better interpret the main results. For example after the first main theorem: “Theorem 2 shows ORF approximate kernel defined by Bessel function of the first kind. This function oscillates contrary to Bessel function of the second kind”. This statement is not actually helping the reader understand the result: a) there is no mention of how Bessel functions behave as a function of the subscript $d$; b) how does this relate with the Gaussian expression that was previously expected?  I believe a concise proof sketch with more detailed intuitive explanations would strongly help the reader.
- Similar considerations as above hold for the appendix. As of now, the appendix consists of a disconnected series of mathematical statements with no linking effort. This strongly penalizes the readability of the text and the quality of the contribution.


# Other comments
- There is no reference for kernel methods, see e.g. [1,2].
- Given that Random Feature models are the main subject of this work, a more extensive coverage of the related work is needed. Indeed, Random Features models have been extensively studied in the learning theory community under the umbrella of the Gaussian Equivalence Principle [3,4,5,6]. These works paved the way for developing a complete theory of the generalization properties of Random Feature models, see e.g. [8]. Analytical approaches based on Random Matrix Theory tools have been developed to study the deep case, see e.g. [9]. As an interesting remark for the authors: [7] analyzes the separability transition for Gaussian and Orthognal Random Features.
- The presented analysis works at finite size. Such a nice aspect should be emphasized more in the manuscript.
- There is a (clearly stated) condition $1<p<d+1$. Why is this assumption necessary? It would be interesting to point out where the proof scheme would fail as in the related RF literature general $(p,d)$ scaling is considered. For example, in the context of learning theory, the generalization properties are identified by m = min(p,d).
- In the introduction/preliminaries please introduce in a few lines Bessel functions for the non-expert reader.


# Minor comments:
- Please rephrase the sentence in the abstract “the former is performed by a random Gaussian matrix and leads exactly to the Gaussian kernel after averaging” .
- Page 2: RHS of equation (2) -> (1)
- Page 2: dxd orthogonal matrix -> pxp orthogonal matrix
- Page 3: please define $\xi$ distributed
- Page 7: scalar product should be denoted by transpose for consistency


# References
- [1] Schölkopf, B., Smola, A. J., Bach, F., et al. Learning with kernels: support vector machines, regularization, optimization, and beyond. 2002.
- [2] Geoffrey S. Watson. Smooth regression analysis.1964.
- [3] Sebastian Goldt, Bruno Loureiro, Galen Reeves, Florent Krzakala, Marc Mezard, and Lenka Zdeborova. The gaussian equivalence of generative models for learning with shallow neural networks 2022.
- [4] Hong Hu and Yue M Lu. Universality laws for high-dimensional learning with random features. 2022.
- [5]  Yatin Dandi, Ludovic Stephan, Florent Krzakala, Bruno Loureiro, and Lenka Zdeborová. Universality laws for gaussian mixtures in generalized linear models. 2024.
- [6] Andrea Montanari and Basil N Saeed. Universality of empirical risk minimization.  2022.
- [7] Generalisation error in learning with random features and the hidden manifold model
F Gerace, B Loureiro, F Krzakala, M Mézard, L Zdeborová. 2020.
- [8] Asymptotics of random feature regression beyond the linear scaling regime. H Hu, YM Lu, T Misiakiewicz. 2024.
- [9] Deterministic equivalent and error universality of deep random features learning
Dominik Schroder, Hugo Cui, Daniil Dmitriev, Bruno Loureiro. 2024.

**Strengths And Weaknesses:**

The main strength of this paper stands in the nice theoretical contribution. The precise bias-variance characterization of the Orthogonal Random Feature (ORF) approximation scheme is a nice technical result. All the main results are presented with coherent numerical illustrations, which is welcomed.

The main weakness of this work is the presentation. In the introduction and global overview of Random Features approximation methods, the relevance of the considered setting is barely discussed; moreover, several works on the topic are missing in the related works description.  Subsequently, the formal presentation of the main results is carried over without any proof sketch and insufficient description of the main ideas, which strongly worsens the readability of the text. Similarly, the appendix consists of a series of disconnected mathematical statements.

I propose in the section below many structural changes that I believe would strongly enhance the presentation of this submission.

---

> ### Author Response · Authors · 2024-06-28
> **Reply to Reviewer ZMow**
>
> We would like to thank the reviewer for the relevant comments and constructive suggestions. We took into account all the suggested improvements to clarify our writing, and in the following discuss the major comments. We have used the colour blue to highlight our answers to these questions.
>
> #### **1. The common belief that ORF approximates the Gaussian kernel**
>
> To avoid confusion, we preferred to remove the sentence in the abstract that emphasizes this point. The confusion comes from the fact that the authors of (Yu et al., 2016) named Orthogonal Random Features the resulting matrix defined as the product of a Haar orthogonal matrix and a random diagonal matrix whose entries are independent $\chi_2$ random variables. Though this model leads to an approximation of the Gaussian kernel, the name may be misleading since the random transformation matrix is not orthogonal due the multiplication by the diagonal matrix. We have clarified that the model we consider in our work is the one denoted ORF’ in Yu et al. (2016), section 4, and is an instance of definition 2.2. in Choromanski et al. (2018) corresponding to one block of features. (page 2)
>
> #### **2. The important role that ORF plays**
>
> RFF are built out of iid random Gaussian matrix. As such, the corresponding model is idealistic. In this sense, ORF may be a more realistic and effective model since it takes into accounts the correlations between samples. ORF was first proposed by Yu et al. (2016), with the goal of approximating the Gaussian kernel. They showed that imposing orthogonality on the randomly generated transformation matrix can reduce the kernel approximation error of RFF when a Gaussian kernel is used. This has led to a number of studies inves- tigating the effectiveness of random orthogonal embeddings and showing empirically and theoretically that ORF estimators achieve more accurate kernel approximation and better prediction accuracy than standard mechanisms based on i.i.d sampling (Choromanski et al., 2017; 2018; 2019). The superiority of orthogonal against random Gaussian projections in learning with random features has also been observed in Gerace et al. (2020). (page 2)
>
> #### **3. Sketch of proof and appendix**
>
> We included after the statement of each result few lines providing a sketch of its proof. The proofs in the appendix are also edited in order to ease their reading.
>
> #### **4. Gaussian Equivalence**
>
> We would like to thank the reviewer for for pointing out these relevant works. We added the references in the introduction. (page 1)
>
> #### **5. The case of $p>d$**
>
> If $p > d$, then the ORF feature map, as introduced in Choromanski et al. (2018), is built out of independent blocks of vectors sampled from independent Haar orthogonal matrices. By linearity and statistical independence, the computations of the bias and of the variance of the ORF estimator follows in this case from summing those we will compute below. (page 3)

---

### Review · Reviewer_LSFZ · 2024-06-07

**Summary Of Contributions:**

The paper presents some theoretical results associated with Random Orthogonal Features based kernel approximation. In particular explicit expectation and variance expressions are derived, showing that contrary to prior belief the expectation is of a Bessel kernel and not a Gaussian kernel. Some exponential bounds on the average ROF kernel are also presented over fairly large range of the distance between its arguments. Some numerical experiments are presented to corroborate the theory.

**Audience:**

Yes

**Broader Impact Concerns:**

Nothing obvious

**Claims And Evidence:**

No

**Requested Changes:**

Can you add some clarity regarding what the reader is to gain from the paper. I appreciate the derivations of the mean and variance of the ORF approximated kernel, but as a reader I do not know what I should be taking away from the paper beyond that. Generally speaking, I would expect most users "want" to approximate the Gaussian kernel due to its popularity. You have shown that the ORF would be biased for this objective, but your MSE experiments don't clarify if the reduction in variance overcomes this bias. I also argue that these do not show a relevant comparison since they have a different "target" (I would argue that most users do not want to approximate the Bessel kernel, and so I question the relevance of the MSE for estimating that target). As it is I see the ORF is biased if I want a Gaussian kernel, but has lower variance. There are also some exponential bounds on the difference between these two kernels over a range which doesn't offer immediate intuitive understanding. The range grows linearly with dimension, as you have shown, but what is relevant to a user is how this range overlaps/not with the range of pairwise distances between observations (after appropriate scaling a la via the use of a bandwidth scaling) which is likely also going to grow with dimension.

**Strengths And Weaknesses:**

Strengths:
- The explicit expectation and variance expressions seem to be novel, and given the common use of ROF these are beneficial results.
- The empirical validation of the results, although they have some issues (see below) are useful support of the theory.

Weaknesses:
- Clarity: Some of the experimental results lack sufficient discussion for total clarity. For example, in relation to Fig. 1, the empirical bias is decreasing with p, and yet the theory says it is independent of p. What should I be concluding from this then? It is also not generally given what value of $z = ||x-y||$ is being used.

- Motivation: Beyond the expectation and variance expressions, I find it hard to see the motivation underlying other parts of what is presented. For example, what is the usefulness of the bounds on the Bessel kernel (the average ROF kernel)? One can see that ROF may be preferable to RFF because it has lower variance, but if one wants to be using a Gaussian kernel and not a Bessel kernel then what should a user choose? This point pertains to the MSE experiment as well. Surely a user of an approximation method, or choosing between two, will have an objective in mind. That is, they may wish to use a Gaussian kernel but are "happy" with an approximate method. Comparing MSE of the two in estimating different things (RFF estimating the Gaussian kernel and ROF estimating the Bessel kernel) doesn't really tell me anything. In relation to this experiment too, why is the Gaussian kernel given a bandwidth and not the Bessel kernel? The scaling of the data within the kernel via a bandwidth is universal to all kernels, so this experiment isn't comparing like with like unless both are given this scaling.

---

> ### Author Response · Authors · 2024-06-28
> **Reply to Reviewer LSFZ (1/2)**
>
> We thank the reviewer for the valuable comments. We have addressed the issues raised by the reviewer, and answer the main ones below point by point. We have used the red colour to highlight our answers to these questions.
>
> > For example, in relation to Fig. 1, the empirical bias is decreasing with p, and yet the theory says it is independent of p. What should I be concluding from this then?
>
> The theory shows that the true expectation of ORF is equal to the Bessel kernel, not the empirical expec- tation. In practice, we approximate the true expectation using an empirical estimation. When p increases, the estimation of the true mean is more accurate. Figure 1 shows that when p is large, i.e. the empirical mean is near to the true mean, the absolute error between the empirical mean and the Bessel kernel function is near to 0. This validates the theory and Theorem 2. The same claim holds for the variance.
>
> > It is also not generally given what value of $z = ||x-y||$ is being used.
>
> In Figure 1, data are generated randomly and the value of $z$ is equal to 24.07. What is relevant in this experiment is having a value of $z$ which is not close to zero (otherwise computations would be trivial) which is the case. Furthermore, since our estimates are valid in an interval whose length grows linearly with the data dimension (of course, our exact formulas for the bias and for the variance are valid everywhere), the values of $z$ may be increased provided $d$ is becomes larger. (page 6)
>
> In Figures 2 and 3, $z$ lies in the $x$-axis. Real-world data are used in Figure 4. The value of $z$ depends on the data described in Table 1.
>
> > What is the usefulness of the bounds on the Bessel kernel (the average ORF kernel)?
>
> RFF are built out of iid random Gaussian matrix. As such, the corresponding model is idealistic.
> In this sense, ORF may be a more realistic and effective model since it takes into accounts the correlations between samples.
> Moreover, the bounds we prove show how far the mean of ORF is from the one of RFF, i.e., how far we are from an iid sampling scheme. The bounds also allows us to identify situations where considering ORF estimator is not far from the RFF one while built out of correlated features (see Figure 2).
>
> On the other hand,  both kernels tend to zero at $z$ tends to infinity. However the Bessel kernel is oscillating and the oscillations become very small after reaching the first zero (of course, the Gaussian kernel is positive and admits a very fast decay to zero). All that to say that our bounds (especially the upper bound) show the closeness  between both kernels before reaching the aforementioned zero.
>
> Let us also stress that a large part of (machine learning) research papers deals with the Gaussian kernel while a very few are concerned with the Bessel one. Accordingly, our paper offers additional results to kernel methods. In particular, we expect our bounds could be useful to transfer theoretical results in learning theory from the RFF to the ORF setting.
>
> > One can see that ORF may be preferable to RFF because it has lower variance, but if one wants to be using a Gaussian kernel and not a Bessel kernel then what should a user choose?
>
> As alluded to above, independence is a strong assumption. Previous studies have reported experimental results showing that some orthogonal random feature models may be preferable to RFF for approximating the Gaussian kernel (see, e.g., Figure 1 in Yu et al. 2016 and Figure 6 in Choromanski et al. 2018). Our theoretical results show that the variance of ORF is less than the one of RFF in an interval whose length grows linearly with the data dimension (see Proposition 6). This guarantees that we can approximate better the corresponding means using ORF than RFF.
>
> It is worth noting that our paper gives new insights on the already proposed and studied ORF model. Actually, it provides the explicit forms of the mean and the variance of ORF, improving our understanding of random orthogonal feature mechanisms in the context of kernel machines.

---

> > ### Comment · Reviewer_LSFZ · 2024-07-08
> > **Responses**
> >
> > Thanks to the authors for their comments and engagements. I have some more to add in reply.
> >
> > 1. The bias decreasing with p: What you are describing seems to indicate a feature of the variability for smaller p, rather than the bias. The theoretical bias should be realisable through simulation/experimentation otherwise this figure doesn't really seem to be showing anything, and it certainly is not validating the theory since it is showing dependence on p where in theory there is no such dependence.
> >
> > 2. Thanks for clarification about the "z" values
> >
> > 3. RFF vs ORF: I am afraid I don't follow the description of RRF as "idealistic" and ORF as "realistic"
> >
> > 4. "Which one to choose?": I am afraid I still do not follow. To what independence are you referring? Also, your comment "This guarantees that we can approximate better the corresponding means using ORF than RFF." relates directly to what I was trying to ask, but does not answer the question. In practice I believe people would like to approximate the Gaussian kernel. Your results show that, if you'll forgive the abuse of notation, E[||ORF - Bessel||^2 ] < E[|| RRF - Gaussian||^2], but this does not answer the question, which is smaller of E[||RRF - Gaussian||^2] and E[||ORF - Gaussian||^2 ]

---

> > > ### Author Response · Authors · 2024-07-18
> > >
> > > We would like to thank the reviewer for his comments.
> > >
> > > 1. Theorem 2 shows that $\mathbb{E}[\tilde{k}_{ORF}(x,y)] = j\_{d/2-1}(||x-y||)$.
> > >
> > > Note that the true mean $\mathbb{E}[\tilde{k}_{ORF}(x,y)] = \int\_{\mathbb{R}^d} \langle \tilde{\phi}(x), \tilde{\phi}(y) \rangle d\mu(w)$.
> > >
> > > The integral being over w. In practice we approximate this integral by sampling. So, a finite number of vector $w_1$, …, $w_p$ will be used to approximate the integral. Figure 1 illustrates the absolute error between Empirical mean of $\tilde{k}_{ORF}(x,y)$ computed using samples $w_1$, …, $w_p$ and Bessel function $j\_{d/2-1}(||x-y||)$. The error is very low for all $p$ from 10 to 300. It varies from $6\*10\^{-3}$ to  $2*10^{-3}$. This shows that our analytic computation is correct and that the  expectation is equal to the Bessel function. We would like to stress that the  error is low even for very small p. The small decrease of p is because  the small scaling of the y-axis and it is justified because we have a  better approximation of the true mean when we use a lager number of  random features.
> > >
> > > 3. RFF is computed using samples $w_1$, …, $w_p$ which are independently and  identically distributed. However the samples in ORF are no more independent since drawn from the Haar measure on the orthogonal group. When $p$ is small, considering that we can approximate well the kernel with a  small number of vectors which are iid is a strong assumption. This is in  some sense the ideal scenario. In ORF, orthogonality allows to better explore the space and when $p$ is small, approximating the kernel in this case is a more realistic scenario. We can modify the formulation and use "more stronger assumption" instead of "idealistic".
> > >
> > > 4. As explained above, the RFF are iid standard Gaussian distribution ORF while ORF are not. If one would like to approximate a Gaussian kernel, our work reveals a bias-variance tradeoff. When using RFF, we approximate better the mean  than ORF but the variance is large. ORF has less variance than RFF but  the bias is larger. The bounds we derived in Proposition 3 give information about how far the mean of ORF is from the gaussian kernel. Moreover, we would like to emphasize that the goal of this work is not about finding the best approximation of the Gaussian kernel. Rather, it focuses on orthogonal random features and gives new insights on this already proposed kernel approximation  technique.

---

> ### Author Response · Authors · 2024-06-28
> **Reply to Reviewer LSFZ (2/2)**
>
> > Comparing MSE of the two in estimating different things (RFF estimating the Gaussian kernel and ROF estimating the Bessel kernel) doesn’t really tell me anything.
>
> The MSE measures the quadratic variability with respect to the empirical data between the estimator and its theoretical mean (the latter is taken with respect to the random features). In other words, the MSE corresponds to an empirical approximation of the variance. Accordingly, Figure 4 supports Proposition 6 since it shows that for the same number of feature p, the approximation error of the kernel function in the ORF setting is smaller than the one in the case of RFF. It gives the average with respect to real-world data of the empirical variance of ORF and RFF estimators. (page 8)
>
> > In relation to this experiment too, why is the Gaussian kernel given a bandwidth and not the Bessel kernel?
>
> For the ORF estimator, introducing such a bandwidth is somehow artificial since the uniform measure on the sphere may be realized as a normalized Gaussian vector (so even if we start with a Gaussian vector whose standard deviation is $\sigma$, this parameter disappears after normalization). Of course, we can consider from the very beginning a Haar orthogonal matrix divided by a positive bandwidth $\sigma$ (which no longer has the significance of a variance as it did in the Gaussian setting). Doing so amounts simply to perform the change $z \mapsto z/\sigma$ in our obtained theoretical results. (page 8)

---

> > ### Comment · Reviewer_LSFZ · 2024-07-08
> > **Continuation**
> >
> > 1. I believe you are mistaken about the definition of the MSE. What you are describing is correct only in an unbiased setting. Mean Squared Error can only be defined in terms of the estimator AND the estimand (target). To reiterate my previous point, if I want to approximate the Gaussian kernel then my MSE is defined in terms of that and not what happens to be the mean of my estimator.
> >
> > 2. Bandwidth: I think perhaps my point is mistaken. How the bandwidth operates is on the scale at which the input space is "measured", and so it would not assign a non-unit standard deviation to the Gaussian random projections (which are then orthonormalised to get the ORF projection) but would apply to the data. That is, the bandwidth should apply as
> > $\tilde K_\sigma(x, y) = p^{-1/2}(sin(w_1'x/\sigma), ..., sin(w_p'x/\sigma), cos(w_1'x/\sigma), ..., cos(w_p'x/\sigma))$

---

> > > ### Author Response · Authors · 2024-07-18
> > >
> > > Continuation:
> > > 1. The expression 'MSE' has a wide sense by its very definition (mean squared error). Of course, one usually measures the quadratic error between a given estimator and the estimated quantity. Nonetheless, we can estimate another errors like the one we displayed in our paper between kernel matrices provided by the theoretical and the empirical means taken at real data. Measuring such an error gives more credit to the robustness of the approximation of the Bessel kernel by orthogonal random features because it takes into account the variability of the data. This is in agreement with previous work, such as Yu et al. (2016), which also used Kernel matrix approximation mean squared error (MSE).
> > >
> > >
> > > 2. This comment is in agreement with our previous answer regarding the bandwidth: we can add in the definition of ORF an extra parameter which will only ''scale" the data, yet it will no longer have the meaning of "variance" as it did  in the study of RFF.

---

### Review · Reviewer_MxJY · 2024-06-21

**Summary Of Contributions:**

The paper proposes an analysis of the bias and variance associated to a special scheme of random features where the features are orthogonal to each other. More precisely:
- Theorem 2 gives the bias of the associated kernel in terms of a normalized Bessel function.
- Proposition 3 gives additional insights about this bias by bounding it by Gaussian kernels.
- Theorem 4 explicits the variance associated to the computation of the approximated kernel.
- Proposition 6 relates this variance to the one associated to the classical RFF kernel.

Numerical experiments complement the theoretical derivations and highlight the main contributions.

**Audience:**

Yes

**Claims And Evidence:**

Yes

**Requested Changes:**

- Clarification of ORF vs structured ORF.
- Clarification on the number of random features.

**Strengths And Weaknesses:**

Strenghts:

- Overall, the paper is very well written and clear.
- Topic is of interest to the TMLR community.
- The authors went to great lengths to explain the findings, giving precious insights beyond the theorems themselves.
- The experimental section is convincing and highlights well the theoretical findings.

Weaknesses:

- The proposed formulation of ORF differs from the original paper and corresponds to what is denoted by structured ORF in the original paper. I have nothing against this choice but it should be mentioned to help the reader.
- The treatment of the relationship between $p$ and $d$ is insufficient in my opinion. It is stated in the notation that $2 \leq p \leq d$ however the paper presents results in expectation over the law of the random features: how useful are such theorems if one cannot take more that $d$ random features ? In the original ORF paper, this was circumvented by taking ORFs from independent sources (hence overall they are not orthogonal anymore), but I do not see it used here.

---

> ### Author Response · Authors · 2024-06-28
> **Reply to Reviewer MxJY**
>
> The authors would like to thank the reviewer for the valuable comments. All the issues pointed out by the reviewer were addressed in the paper. We have colour-coded our answers with purple.
>
> #### **1. ORF vs structured ORF**
> The model we consider here is the one denoted ORF’ in Yu et al. (2016), section 4, and is an instance of definition 2.2. in Choromanski et al. (2018) corresponding to one block of features. (see page 2)
>
> To speed up the computations, structured ORF makes use of structured orthogonal matrices, which is not the case in this work. SORF is out of the scope of this paper.
>
> #### **2. The number of random features $p$**
> If $p > d$, then the ORF feature map, as introduced in Choromanski et al. (2018), is built out of independent blocks of vectors sampled from independent Haar orthogonal matrices. By linearity and statistical independence, the computations of the bias and of the variance of the ORF estimator follows in this case from summing those we will compute below. (see pages 2-3)

---

### Author Response · Authors · 2024-06-28
**Reply to Reviewers’ Comments**

We explain below the changes that have been made to the paper “Orthogonal Random Features: Explicit Forms and Sharp Inequalities” after the initial reviews. Before going to the details, the authors would like to sincerely thank the anonymous reviewers for their careful reading of the paper, and for the accurate comments and recommendations that helped us to improve the overall quality of our contribution. In our revised version, in order to facilitate reading, we have used colour-coding in highlighting our new additions to the paper. The answers to each of the reviewers have their own colour (Rev. MxJY: purple, Rev. LSFZ: red, Rev. ZMow: blue).

---

### Author Response · Authors · 2024-09-08
**Camera-Ready Version**

The camera-ready version is now uploaded. We would like to thank the reviewers and the action editor for their constructive comments and suggestions that improved the presentation of the paper.

---

### Decision · Action_Editor_Y6iF · 2024-08-17

**Recommendation:** Accept with minor revision

**Comment:**

The theoretical results are sound and interesting, and for this reason I am recommending the paper for acceptance at TMLR.

There were some divergence points in the discussion with Reviewer LSFZ concerning: (a) How meaningful is using the MSE to compare RFF and ORF, e.g. in Fig. 4; (b) Whether ORF is more "realistic" than RFF".

After a careful reading of this discussion and of the revised version, I believe point (a) was addressed. There might be some discussion on what conclusions can be taken from this comparison, but at least the authors have clarified the rationale.

However, I am on the side of Reviewer LSFZ on point (b).  What is "realistic" in the design of an algorithm or approximation scheme is subjective and unclear. I believe the word "effective" that follows already conveys a more precise meaning in terms of the theoretical results for the variance. Since this word is not really adding anything to the discussion, I request the authors to remove it from the camera-ready version.

**Audience:**

The reviewers agree the subject of this work are of interest to the TMLR community interested on theoretical aspects of kernel methods.

**Claims And Evidence:**

This work investigates the advantage of orthogonal random features with respect to Gaussian i.i.d. random featutes when approximating a kernel method. Its main results are: (a) an expression for the bias-variance decomposition for the kernel approximation (Theorem 1); (b) an explicit expression for bias of the approximation. The authors also provide some numerical experiments comparing these two approximations in real tasks.

The theoretical results are rigorous and neat. However, as it came out in the discussion with Reviewer LSFZ, there is some room for confusion between what is proven and the discussion on the discussion for the relevance to the practice of random features regression.